# Antioxidant-Rich *Clitoria ternatea* Flower Extract Promotes Proliferation and Migration of Human Corneal Epithelial Cells

**DOI:** 10.3390/plants14203216

**Published:** 2025-10-20

**Authors:** Karthini Devi Rajan, Nahdia Afiifah Abdul Jalil, Taty Anna Kamarudin, Fairus Ahmad

**Affiliations:** Department of Anatomy, Faculty of Medicine, Universiti Kebangsaan Malaysia (UKM), Jalan Yaacob Latif, Bandar Tun Razak, Kuala Lumpur 56000, Malaysia; karthinidevi92@gmail.com (K.D.R.); nahdia.afiifah@ukm.edu.my (N.A.A.J.); tatykamarudin@ukm.edu.my (T.A.K.)

**Keywords:** *Clitoria ternatea*, corneal epithelial cells, antioxidant activity, phytochemicals, cell migration

## Abstract

A corneal abrasion results from the disruption or loss of cells in the corneal epithelium. If inadequately treated, it can compromise visual clarity. The wound healing process of a corneal abrasion involves epithelial migration, proliferation and adhesion. *Clitoria ternatea* flower extract (CTE) is rich in flavonoids, anthocyanins and other bioactive compounds. It has antioxidant, anti-inflammatory and wound-healing properties. This study explores the potential of CTE to be used as a natural supplement to improve corneal wound healing. Phytochemical profiling via LC–MS identified a total of 51 distinct bioactive constituents. The anthocyanin content, quantified in terms of cyanidin-3-glucoside equivalent, was quantified at 33.06 mg per gram of extract. The extract exhibited 33.8% DPPH radical scavenging activity and a total polyphenol content equivalent to 24.14 mg/g gallic acid. Human telomerase-immortalized corneal epithelial (hTCEpi) cells maintained in keratinocyte basal medium were utilized to determine cytotoxicity and wound-healing effects. The optimal extract concentration of 0.08 mg/mL, quantified via MTT assay, resulting in enhanced cell viability. Scratch assays demonstrated a higher percentage of wound closure in the CTE-treated group at 6 and 12 h relative to the untreated group, with statistical significance (*p* < 0.05). The gene expressions of CK3 and Cx43, quantified via qRT-PCR, showed no significant differences between groups. However, within the CTE-treated group, CK3 expression increased at 12 h relative to 0 h and 6 h, and Cx43 expression rose significantly at 12 h compared with 0 h (*p* < 0.05). Immunofluorescence confirmed positive protein expression of both markers. These findings suggest that CTE possesses potent antioxidant properties and promotes corneal epithelial wound healing through upregulation of CK3 and Cx43 in vitro.

## 1. Introduction

The cornea is the eye’s primary refractive surface, comprising five layers that ensure optical clarity and serve as a protective barrier [1]. Limbal epithelial stem cells, located at the corneoscleral junction, enable continuous regeneration of the epithelium to maintain corneal transparency [2].

Corneal abrasion is a common ocular injury caused by trauma, foreign bodies, and improper contact lens use [3]. Patients often present with pain, redness, tearing, and photophobia, and central abrasions can impair vision [4]. Untreated abrasions may progress to ulcers, microbial keratitis, or stromal scarring, leading to permanent visual impairment [5]. In Malaysia and other Southeast Asian countries, occupational and environmental risks contribute to the high burden of corneal abrasions [6].

Conventional management includes lubricants, topical antibiotics, and analgesics; however, current therapies are limited, and adjunctive treatments that accelerate epithelial recovery are still needed [7]. Natural products such as omega-3 fatty acids and Manuka honey have demonstrated potential to improve wound healing outcomes [8]. For instance, 0.025% Acacia honey promotes the proliferation of corneal epithelial cells while maintaining their normal phenotype and enhancing the expression of key wound healing proteins [9]. Similarly, *Centella asiatica* aqueous extract has been found to stimulate cell proliferation and migration of corneal epithelial cells, suggesting its role in accelerating wound healing [10].

At the molecular level, cytokeratin 3 (CK3) and connexin 43 (Cx43) are important markers of corneal epithelial homeostasis and wound healing. CK3 contributes to epithelial integrity [11], while Cx43 regulates intercellular communication during tissue repair [12].

*Clitoria ternatea* (butterfly pea flower) is rich in anthocyanins, flavonoids, and cyclotides that exhibit antioxidant, anti-inflammatory, and antimicrobial activities [13]. Prior studies have shown its wound-healing potential in diabetic animal models [7]. However, little is known about its role in corneal epithelial biology, particularly its effects on corneal epithelial cell proliferation, migration, and wound healing. This study therefore investigates the ability of CTE to enhance corneal epithelial cell proliferation, migration, and wound healing. By linking its phytochemical profile to cellular outcomes, this work addresses a critical gap in the development of natural adjuvant therapies for corneal injury.

## 2. Results

### 2.1. Phytochemical Analysis of CTE

#### 2.1.1. Phytochemical Profiling

LC-MS analysis revealed the presence of various phytochemical constituents in CTE. The total ion chromatogram (TIC) showed distinct peak profiles under both positive and negative ionization modes (Figure 1). The LC-MS analysis revealed 14 peaks in the chromatogram acquired under positive ionization mode and 8 peaks under negative ionization mode, indicating the presence of multiple phytochemical constituents in the CTE. A total of 51 phytochemicals were tentatively identified based on matching retention times and *m*/*z* values with the LC-MS database. The predicted compounds were annotated using the METLIN metabolomics database (Agilent Technologies, Santa Clara, CA, USA) in combination with Agilent Mass Hunter Qualitative Analysis B.05.00 software. Compound identification was based on accurate mass matching and should therefore be considered tentative in the absence of authentic reference standards or fragmentation analysis.

The identified compounds include flavonoids, anthocyanins, fatty acids, alkaloids, terpenoids, steroids, and other antioxidant-related groups. Notable examples include delphinidin, kaempferol derivatives, luteolin glycosides, and aurantinidin. Although an aqueous extraction was used, the presence of lipophilic compounds such as fatty acids and terpenoids may be explained by their partial solubility at elevated temperatures (50 °C) or by co-extraction of trace amounts. This possibility has been acknowledged as a limitation. Table 1 summarizes the antioxidant group and its representative compounds. The complete list of 51 phytochemicals is provided in Appendix A.

#### 2.1.2. Anthocyanin Content

The anthocyanin concentration, measured as cyanidin-3-glucoside equivalent, was 33.06 mg/g of extract, equivalent to 3.3% anthocyanin content in CTE.

#### 2.1.3. Antioxidant Activity (DPPH Assay)

The DPPH radical scavenging activity of CTE was 33.8% at 0.1 g/mL, which is lower than that of the synthetic antioxidant BHA (95%) and the natural standard ascorbic acid (92%) at equivalent concentrations. Based on commonly used classifications, this level of scavenging activity is considered modest.

#### 2.1.4. Total Polyphenol Content

By the Folin–Ciocalteu method, the total phenolic content was quantified at 24.14 mg gallic acid equivalents (GAE) per gram of extract.

### 2.2. Determination of Optimal Dose for Cell Viability

The MTT assay revealed that 0.08 mg/mL showed the highest cell viability (183.34 ± 3.70%) within the 72 h timeframe tested. Statistically significant increases in viability were observed at 0.07 and 0.08 mg/mL compared with lower and higher concentrations (a and b, *p* < 0.05; Figure 2). This optimal dose was used in all subsequent experiments.

### 2.3. Effect of CTE on hTCEpi Cell Migration

The wound healing model of cell scratch assay demonstrated that treatment with 0.08 mg/mL CTE significantly enhanced wound closure compared to control. Complete wound closure (100%) was observed after 12 h in the treated group (F), whereas the untreated group achieved only 51.83 ± 1.11% closure (C) (a, *p* < 0.05) (Figure 3 and Figure 4). At 6 h, the treated group showed 67.50 ± 1.58% wound closure, while the untreated group exhibited 33.67 ± 0.71% closure (a, *p* < 0.05) (Figure 4). Additionally, wound closure increased significantly over time at 0, 6, and 12 h within both the CTE-treated group (b and d, *p* < 0.05) and the groups that are not treated (c and e, *p* < 0.05) (Figure 4).

### 2.4. Gene Expression Analysis (qRT-PCR)

qRT-PCR analysis indicated no statistically significant differences between treated and untreated groups for CK3 or Cx43 at equivalent time points. However, within the treated group, CK3 expression was markedly higher at 12 h compared with both 0 and 6 h (a, *p* < 0.05). In untreated cultures, CK3 levels also increased significantly at 12 h relative to baseline (b, *p* < 0.05). For Cx43, expression rose significantly in treated cells at 12 h versus 0 h (a, *p* < 0.05) (Figure 5 and Figure 6).

### 2.5. Protein Expression via Immunofluorescence

Immunofluorescence staining confirmed expression of CK3 and Cx43 proteins in hTCEpi cells. CK3 was localized in the cytoplasm and membrane (green), while Cx43 appeared as red fluorescence in the cytoplasm. Nuclear staining with DAPI (blue) verified cellular morphology. Cells treated with 0.08 mg/mL CTE showed positive protein expression of CK3 and Cx43 at the wound margins (D and E) and within the wound at 12 h (F) post-treatment (Figure 7).

## 3. Discussion

The LC–MS analysis confirmed that CTE contains diverse bioactive compounds, including flavonoids, anthocyanins, alkaloids, fatty acids, terpenoids, phospholipids, and vitamin D analogs [14,15]. Flavonoids such as kaempferol and luteolin are known to reduce oxidative stress and modulate inflammatory pathways, both of which are implicated in epithelial proliferation and repair [16]. Anthocyanins, including cyanidin and delphinidin, can stimulate collagen synthesis and cell migration, processes directly relevant to wound closure in corneal epithelium [17]. These bioactive compounds may exert their effects through a combination of antioxidant, anti-inflammatory, and cytoprotective mechanisms, all of which are critical for promoting tissue regeneration in the cornea. Fatty acids and alkaloids may provide additional antimicrobial and anti-inflammatory support that could contribute to corneal defense and regeneration [18]. It should be noted that the detection of lipophilic compounds from an aqueous extract may reflect partial solubility at higher extraction temperatures or minor co-extraction. This highlights a methodological limitation and should be explored in future extraction optimization studies.

Interestingly, the anthocyanin content obtained in this study (33.06 mg/g) was higher than previously reported [19]. This difference may be due to the use of aqueous extraction, which enhances compound yield compared to solvent-based methods [20]. Given the role of anthocyanins in antioxidant defense and tissue remodeling, higher yields may increase the biological potency of CTE for ocular applications [21]. The presence of these compounds suggests that CTE may help mitigate oxidative damage, modulate inflammation, and stimulate wound healing in the corneal epithelium, making it a promising candidate for ocular injury treatment [22].

The DPPH assay demonstrated that CTE possessed a scavenging capacity of 33.8%, which is modest compared with synthetic antioxidants such as BHA and natural standards such as ascorbic acid [23]. Nevertheless, even moderate antioxidant activity may be physiologically meaningful, as oxidative stress is known to impair corneal epithelial wound healing [24]. In the context of corneal injury, reducing oxidative damage, even modestly, could contribute to a more favorable healing environment. Moreover, antioxidant assays alone do not capture the synergistic activity of phytochemicals; thus, the observed enhancement of proliferation and migration in vitro may result from combined effects of multiple compounds present in CTE, such as flavonoids, anthocyanins, and fatty acids, which collectively enhance cellular responses to injury through antioxidative, anti-inflammatory, and proliferative pathways [21].

The MTT assay showed that 0.08 mg/mL produced the highest cell viability within the 72 h timeframe tested. However, this concentration should not be considered “optimal” without further validation, as the assay used a narrow concentration range and a single readout. Such biphasic effects are common with polyphenols, where lower concentrations stimulate proliferation, but higher concentrations may induce cytotoxicity [20]. Future studies should extend the dose–response range and include positive controls such as epidermal growth factor (EGF) for proliferation or hydrogen peroxide (H_2_O_2_) for cytotoxicity [25].

The wound-healing assay confirmed faster closure in CTE-treated cells, consistent with enhanced proliferation and migration. CK3 expression increased at 12 h, reflecting its role in epithelial differentiation during healing [22], while Cx43 expression was also upregulated, indicating active intercellular signaling at the wound margin [23]. Similar effects have been described with other natural agents, including edible bird’s nest and bee venom [26,27]. The observed increase in Cx43 expression may be consistent with facilitation of tissue repair [28]. The upregulation of Cx43 in corneal fibroblasts via insulin-like growth factor-1 (IGF-1) facilitates cell proliferation, migration, and differentiation during wound healing by enabling intercellular signaling such as MAPK/ERK pathway. This communication helps coordinate the healing process, ensuring proper regeneration of the epithelium [29]. Elevated Cx43 in later stages supports extracellular matrix remodeling, fibroblast activation, and angiogenesis, all critical for reinforcing and stabilizing healed corneal tissue [30].

Despite the absence of a positive control such as growth factor, the results from the scratch assay still provide meaningful insights as the treated group demonstrated significant cell proliferation and migration, indicating a clear improvement in wound healing. Although CK3 and Cx43 expression levels appeared elevated at 12 h in the CTE-treated group, these differences were not statistically significant compared with the untreated controls, which also showed increased expression over time. Therefore, the present findings cannot confirm a direct effect of CTE on these molecular markers. While the trends observed may be consistent with known roles of CK3 in epithelial differentiation and Cx43 in intercellular communication, further studies incorporating positive controls, additional replicates, and in vivo validation are required to determine whether CTE exerts a true modulatory effect on these proteins. In addition, between 0 h and 6 h, the treatment group showed a significantly higher rate of CK3 and Cx43 expression compared to the control group, suggesting that the treatment stimulates a stronger or more rapid differentiation response during the early stages of corneal wound healing.

This study suggests that CTE holds potential as a therapeutic agent for corneal epithelial injuries. Its bioactive compounds such as flavonoids, anthocyanins, and alkaloids are known to regulate oxidative stress, inflammation, and cellular proliferation, which are critical for wound healing [31]. The observed improvement in cell proliferation, migration, and wound closure indicates that CTE may promote corneal regeneration, making it a potential adjunct in treating corneal injuries or disorders.

To translate these findings into clinical practice, further studies are necessary to optimize the formulation and delivery of CTE, potentially through eye drops or gels, to ensure effective application to the ocular surface. Additionally, in vivo studies using animal models are needed to evaluate the safety and efficacy of CTE in a more complex biological context. Incorporating positive controls such as EGF and investigating its effects on molecular markers like CK3 and Cx43 will be essential for confirming its therapeutic potential. These studies, combined with further exploration of its molecular mechanisms, will facilitate the development of CTE as a viable natural therapy for corneal epithelial injury and other ocular conditions.

## 4. Limitations

This study has several limitations that should be acknowledged. First is the absence of detailed cultivation site information for the plant material. Although authentication was performed and a voucher specimen deposited, future studies should ensure full traceability of plant sources, including cultivation site and geographic coordinates. The phytochemical profile through METLIN database was performed without authentic standards or MS/MS fragmentation should therefore be regarded as preliminary, and future studies should include reference standards and quantitative analyses to confirm compound identities and relative abundance.

The MTT assay was conducted within a relatively narrow concentration range (0.01–0.1 mg/mL). Although this range was selected based on preliminary pilot studies and solubility constraints, a broader dose–response analysis and additional time points are needed to confirm whether 0.08 mg/mL represents an optimal concentration or simply the most effective under the conditions tested.

There are no positive controls included in the assays. In particular, epidermal growth factor (EGF) could serve as a proliferation and wound-healing benchmark, while hydrogen peroxide (H_2_O_2_) could act as a cytotoxic comparator.

Similarly, the scratch assay compared only untreated and CTE-treated cells, without a positive healing control. The absence of these controls limits direct comparison with established modulators of corneal wound healing.

Finally, the findings are restricted to an immortalized corneal epithelial cell line in an in vitro setting. Future studies should therefore incorporate positive controls, primary human corneal epithelial cells, in vivo validation, and comparison with standard therapeutic agents to strengthen the translational relevance of CTE.

## 5. Materials and Methods

Ethical approval for this research was obtained from the Universiti Kebangsaan Malaysia Research Ethics Committee (Reference: JEP-2023-563) on 24 August 2023. All experimental work complied with the institutional guidelines for ethical conduct in research.

### 5.1. Plant Material and Extract Preparation

Fresh flowers of *Clitoria ternatea* were purchased from a supplier in Petaling Jaya, Malaysia. Precise cultivation site details and geographic coordinates were not available from the supplier. To ensure accurate identification, the plant material was authenticated by the Herbarium, Universiti Kebangsaan Malaysia (UKMB), and a voucher specimen was deposited under the accession number ID081/2023. The flowers were processed using an aqueous extraction method, where they were blended with DNA/RNA-free distilled water (Invitrogen, Thermo Fisher Scientific, Waltham, Massachusetts, USA) in a 1:15 (*w*/*v*) proportion. The mixture was incubated at 50 °C for 30 min, filtered, freeze-dried, and stored at 8 °C. A stock solution was prepared at a concentration of 10 mg/mL.

### 5.2. Phytochemical Analysis of CTE

#### 5.2.1. Phytochemical Profiling

LC–MS analysis was carried out using an ACQUITY I-Class Ultra-High-Performance Liquid Chromatography (UHPLC) system (Waters Corporation, Milford, Massachusetts, USA) equipped with a binary pump, autosampler, vacuum degasser, and column oven. Separation of phenolic compounds was achieved on an ACQUITY UPLC HSS T3 column (100 mm × 2.1 mm × 1.8 µm) maintained at 40 °C. The mobile phases consisted of 0.1% formic acid in water (A) and acetonitrile (B), with a binary linear gradient at a flow rate of 0.6 mL/min. The injection volume was set to 1 µL. The UHPLC system was connected to a Vion IMS QTOF hybrid mass spectrometer (Waters Corporation, Milford, Massachusetts, USA) using a LockSpray ion source in positive and negative electrospray ionization modes. LC–MS analysis was performed using both positive (ESI+) and negative (ESI−) electrospray ionization modes to ensure broader metabolite coverage. Parameters included a capillary voltage of 1.50 kV, reference capillary voltage of 3.00 kV, source temperature of 120 °C, desolvation temperature of 550 °C, desolvation gas flow at 800 L/h, and cone gas flow at 50 L/h. Nitrogen (>99.5%) was used for both desolvation and cone gases. Data acquisition was performed in high-definition MSE (HDMSE) mode over an *m*/*z* range of 50–1500 with a scan interval of 0.1 s.

#### 5.2.2. Anthocyanin Content

Anthocyanin levels were quantified at the UKM-MTDC Technology Centre, Universiti Kebangsaan Malaysia, using an HPLC system (Thermo Fisher Scientific, Waltham, Massachusetts, USA) following a modified method [14]. The chromatographic column was identical to that used for LC–MS. The content was expressed as cyanidin-3-glucoside equivalents (mg/g) by comparison with a standard. Two mobile phases were prepared: Solvent A—trifluoroacetic acid in deionized water (pH 2.5) and Solvent B—100% methanol. A gradient program was applied as follows: 0 min, 100% A; 20 min, 50% A/50% B; 30 min, 100% B; and 35–40 min, 100% A. Detection was set at 280 nm with a 20 µL injection volume. Extraction involved hydrolyzing 1 g of CTE with a mixture of 12 mL methanol, 8 mL deionized water, and 5 mL 6 M HCl at 95 °C for 2 h, followed by cooling, filtration through a 0.45 µm Whatman nylon membrane (Waters GmbH, Eschborn, Hessen, Germany ), and analysis.

#### 5.2.3. Antioxidant Activity (DPPH Assay)

The antioxidant potential of CTE was measured using the DPPH radical scavenging method at the UKM-MTDC Technology Centre [32]. A 100 µL aliquot of extract at concentrations between 200 and 1000 µg/mL was mixed with 3.9 mL DPPH solution (25 mg/L in aqueous medium) and kept in the dark for 30 min. Absorbance was recorded at 515 nm. Methanol served as a blank, the positive control consisted of methanol with DPPH, and butylated hydroxyanisole (BHA) was used as the reference antioxidant. Results were expressed as the percentage of DPPH radical inhibition using:

% Inhibition = [(Absorbance_Control_ − Absorbance _Sample_) ÷ Absorbance _Control_] × 100

#### 5.2.4. Total Polyphenol Content

Polyphenol content was determined via the Folin–Ciocalteu assay [33]. In brief, 100 µL ethanolic extract (1 mg/mL) was combined with 7.9 mL distilled water and 0.5 mL Folin–Ciocalteu reagent. After 2 min, 1.5 mL 7.5% sodium carbonate was added, and the mixture was incubated for 2 h at room temperature. Absorbance was measured at 765 nm (Nakagyo-ku, Kyoto, Japan). Values were expressed as gallic acid equivalents (mg GAE/g dry extract).

### 5.3. Cell Culture

The hTCEpi cell line was expanded, which we obtained from Evercyte. Human telomerase-immortalized corneal epithelial (hTCEpi) cells were maintained under sterile culture conditions. Cells stored in liquid nitrogen (−196 °C) were thawed rapidly in a 37 °C water bath and transferred to a 15 mL centrifuge tube containing 9 mL of Keratinocyte Basal Medium (KBM; Lonza, Morristown, New Jersey, USA). Following centrifugation at 170× *g* for 5 min at room temperature, the resulting pellet was gently resuspended in 1 mL of fresh KBM. The cell suspension was seeded into T25 flasks pre-equilibrated with 3 mL KBM. Cultures were incubated at 37 °C in 5% CO_2_, with the first medium change after 24 h and subsequently every 48 h. When cells reached 70–80% confluence, subculturing was performed using 0.05% EDTA. Flasks were rinsed with phosphate-buffered saline (PBS) and treated with 2 mL EDTA for 2.5 min at 37 °C to detach cells, followed by gentle tapping. A second detachment step with another 2 mL EDTA for 1.5 min was performed to ensure complete release. Cells were pooled, centrifuged at 170× *g* for 5 min, the supernatant discarded, and the pellet resuspended in 4 mL KBM. Viability was assessed via trypan blue exclusion (90 µL cell suspension mixed with 10 µL trypan blue), and viable cells were counted using a hemocytometer under an inverted microscope. Passages up to 11 were used for downstream experiments, including migration assays, gene expression analysis, and immunofluorescence.

### 5.4. MTT Assay for Cytotoxicity and Optimal Dose

For cytotoxicity evaluation and dose optimization, hTCEpi cells (5 × 10^4^ cells/well) were seeded into 96-well plates and treated with CTE at concentrations ranging from 0.01 to 0.1 mg/mL. After 72 h, 10 µL of MTT reagent (5 mg/mL) was added to each well and incubated for 4 h. Formazan crystals were dissolved using DMSO, and absorbance was measured at 570 nm. The concentration showing the highest cell viability was selected for subsequent experiments. This assay was limited by the use of a relatively narrow concentration range (0.01–0.1 mg/mL). Although this range was selected based on preliminary pilot studies and solubility constraints, a broader dose–response analysis would provide a more comprehensive understanding of the cytotoxic and proliferative effects of CTE. In addition, positive controls such as epidermal growth factor (EGF) for proliferation or hydrogen peroxide (H_2_O_2_) for cytotoxicity were not included. The absence of these controls represents a limitation of the experimental design and should be addressed in future studies.

### 5.5. Cell Scratch Assay

Cell scratch assays were performed on confluent hTCEpi monolayers. A vertical scratch was made using a 200 µL pipette tip. Cells were divided into control and CTE treatment groups. Wound healing was imaged at 0, 6, and 12 h. Wound closure (%) was calculated using the formula: [G(0) − G(t)]/G(0) × 100%.

### 5.6. RNA Extraction and qRT-PCR

Total RNA was isolated from cells using TRI reagent, and concentration and purity were assessed with a NanoDrop spectrophotometer. Complementary DNA (cDNA) synthesis was performed with LunaScript RT SuperMix (New England Biolabs, Ipswich, Massachusetts, USA). Quantitative real-time PCR (qRT-PCR) was carried out using Luna Universal qPCR Master Mix with primers specific for CK3, Cx43, and GAPDH. Gene expression was calculated using the 2^ΔCt^ method.

### 5.7. Immunofluorescence Staining

Fixation of cells carried out with 4% paraformaldehyde, permeabilized and blocked. Primary antibodies against CK3 and Cx43 were applied, followed by fluorescent secondary antibodies. Nuclei were counterstained with DAPI. Fluorescent images were obtained using a fluorescence microscope.

### 5.8. Statistical Analysis

Statistical analyses were conducted using GraphPad Prism 9 software. Parametric datasets were analyzed using two-way ANOVA followed by Tukey’s post hoc test, while non-parametric data were evaluated using the Kruskal–Wallis test and Mann–Whitney U test. Results are presented as mean ± standard error of the mean (SEM), with *p*-values < 0.05 considered statistically significant.

## 6. Conclusions

This study demonstrates that *Clitoria ternatea* flower extract (CTE) enhances corneal epithelial cell proliferation and migration in vitro and modulates CK3 and Cx43 expression, suggesting a potential role in supporting epithelial repair. The phytochemical analysis confirmed the presence of bioactive compounds, including flavonoids and anthocyanins, which may contribute to these observed effects through antioxidant and wound-healing mechanisms.

However, these findings remain preliminary and are limited to an immortalized corneal epithelial cell line without the inclusion of positive controls or in vivo validation. Future studies should incorporate primary human corneal cells, appropriate proliferation and wound-healing controls, and animal models to confirm the therapeutic relevance of CTE. Comparative studies with standard therapeutic agents will also be necessary before considering its clinical translation as an adjunctive treatment for corneal injury.

## Figures and Tables

**Figure 1 plants-14-03216-f001:**
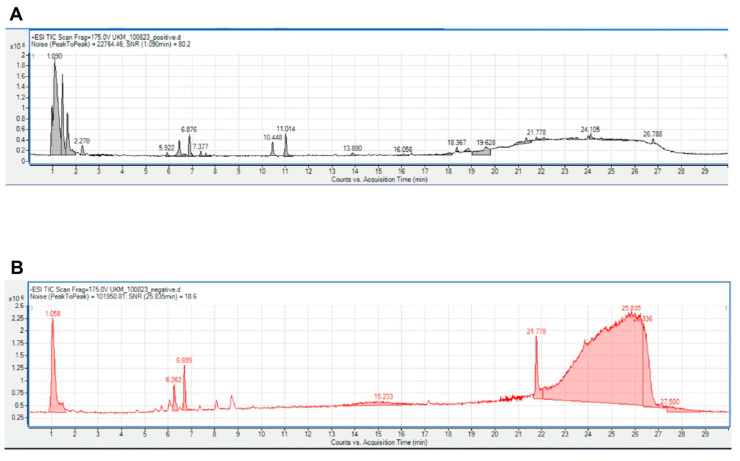
The retention times of the compound peaks produced by chromatography of CTE flower extract. The retention times of compounds formed in positive (**A**) and negative (**B**) ionization modes are shown in the figure above.

**Figure 2 plants-14-03216-f002:**
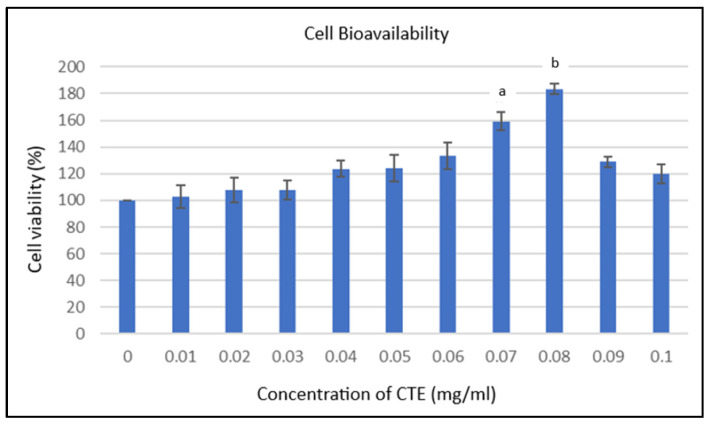
Percentage of hTCEpi cell viability on day three of culture. hTCEpi cells were treated with serial dilutions of CTE at graded concentrations (0.01–0.1 mg/mL). Values represent mean ± standard error of the mean (S.E.M), *n* = 6. Statistical analysis was performed using two-way ANOVA followed by Tukey’s post hoc test. a indicates a significant difference (*p* < 0.05) compared with treatment concentrations from 0 to 0.05 mg/mL and 0.1 mg/mL. b Indicates significant compared to treatment doses from 0 to 0.06 mg/mL and 0.09 to 0.1 mg/mL of *Clitoria ternatea* extract (*p* < 0.05).

**Figure 3 plants-14-03216-f003:**
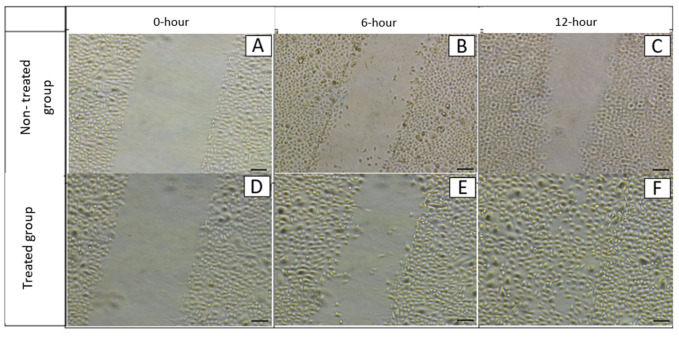
Photomicrographs showing the wound area and hTCEpi cell migration captured using time-lapse microscopy at the beginning of the experiment (0-h), after 6 h (6-h), and after 12 h (12-h), with *n* = 6. The in vitro hTCEpi wound model without treatment is shown in photomicrographs (**A**–**C**), while the model treated with 0.08 mg/mL CTE flower extract is shown in photomicrographs (**D**–**F**). Magnification: ×10. Scale bar = 100 µm.

**Figure 4 plants-14-03216-f004:**
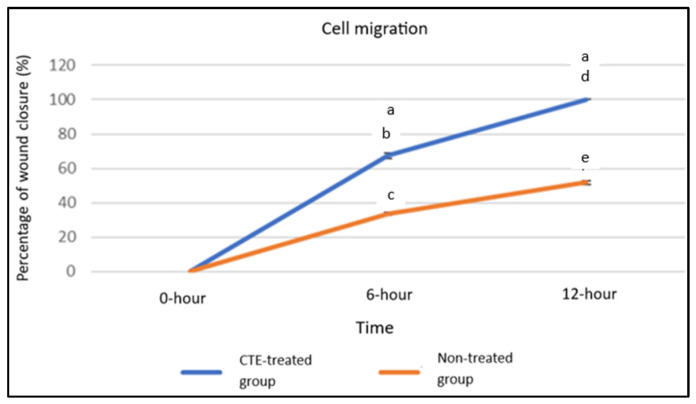
Percentage of wound closure in the hTCEpi cell migration assay for cultures without treatment and with 0.08 mg/mL CTE flower extract treatment. Data were analyzed using two-way ANOVA followed by Tukey’s post hoc test and are expressed as mean ± standard error of the mean (S.E.M), with *n* = 6. a: Significant compared to untreated hTCEpi cells at 6 and 12 h (*p* < 0.05). b: Significant compared to treated hTCEpi cells at 0 h. c: Significant compared to untreated hTCEpi cells at 0 h. d: Significant compared to treated hTCEpi cells at 0 and 6 h. e: Significant compared to untreated hTCEpi cells at 0 and 6 h.

**Figure 5 plants-14-03216-f005:**
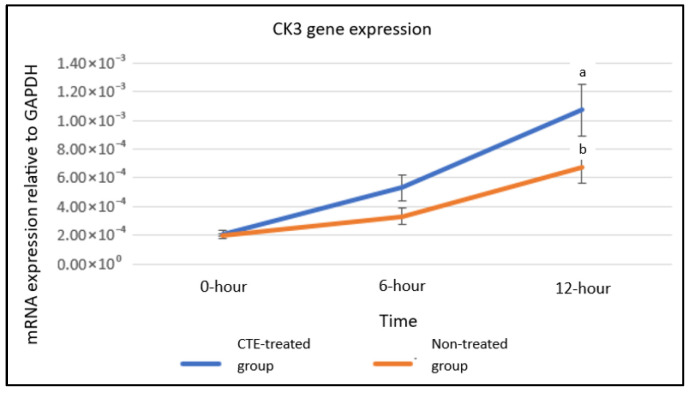
CK3 gene expression in hTCEpi cells cultured without treatment and with 0.08 mg/mL CTE flower extract treatment. Data were analyzed using two-way ANOVA followed by Tukey’s post hoc test and are expressed as mean ± standard error of the mean (S.E.M), with *n* = 6. a: Significant compared to hTCEpi cells treated with 0.08 mg/mL CTE at 0 and 6 h (*p* < 0.05). b: Significant compared to untreated hTCEpi cells at 0 h (*p* < 0.05).

**Figure 6 plants-14-03216-f006:**
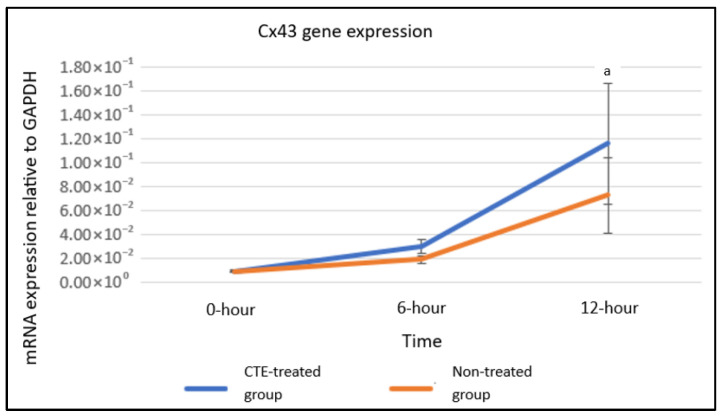
Cx43 gene expression in hTCEpi cells cultured without treatment and with 0.08 mg/mL CTE flower extract treatment. Data were analyzed using two-way ANOVA and Tukey’s post hoc test, and results are presented as mean ± standard error of the mean (S.E.M), where *n* = 6. a: significant compared to hTCEpi cells treated with 0.08 mg/mL CTE at 0 h (*p* < 0.05).

**Figure 7 plants-14-03216-f007:**
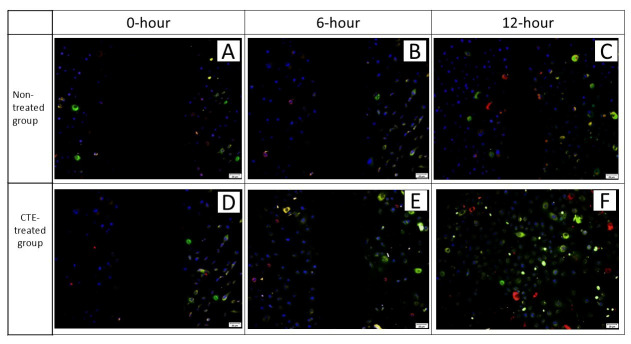
Photomicrographs showing immunofluorescence staining of CK3 (green) and Cx43 (red) proteins at the wound margins in vitro, representing one-sixth of the full hTCEpi cell monolayer at 0 h, 6 h, and 12 h. CK3 protein (green) is indicated by green-stained cytoplasm, while Cx43 protein (red) is indicated by red-stained membranes and cytoplasm. Cell nuclei (blue) are stained in blue. Cells were non-treated (**A**–**C**) and treated with 0.08 mg/mL CTE (**D**–**F**). Magnification: ×100. Scale bar = 100 µm.

**Table 1 plants-14-03216-t001:** Quantification of phenolics, flavonoids, and antioxidant properties in CTE flower extract.

Group	Number of Compounds	Representative Examples
Flavonoids	7	Kaempferol, Luteolin, Scutellarein
Anthocyanins	6	Delphinidin, Pelargonidin, Cyanidin
Fatty acids	7	9(S)-HpODE, Hydroxy-octadecanoic acids
Alkaloids	3	Vernoflexuoside, Isokarbostiril
Terpenoids	2	Tiarubrin A, Ectocarpen
Peptides	1	Thr-Phe
Others	10+	Includes vitamin analogs, phosphates, prostaglandins

## Data Availability

The original contributions presented in this study are included in the article. Further inquiries can be directed to the corresponding author(s).

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
