# Peer review of "Antioxidant-Rich *Clitoria ternatea* Flower Extract Promotes Proliferation and Migration of Human Corneal Epithelial Cells"

_plants, 2025, doi:10.3390/plants14203216_

Round 1

Reviewer 1 Report

Comments and Suggestions for Authors

The authors presented in vitro study of the effects of Clitoria ternatea flower extract on the corneal epithelial cell proliferation, migration, and wound healing. The topic is relevant and of interest for the readers. The manuscript is generally well written, with clear methodology and adequate discussion. However, some sections require correction, additional context, and interpretation.
I would recommend expanding the introduction in order to clarify how the present findings advance current knowledge compared with other natural agents.
Remove the excessive epidemiological details and focuse more on the knowledge gap.
The LC–MS profiling identified 51 compounds, but the discussion does not adequately connect specific compounds to wound healing activity.
About MTT assay: Could higher doses have led to saturation or cytotoxic effects?
The reported antioxidant activity is modest compared to standard antioxidants. Include in the discussion interpretation about this antioxidant activity. Is it meaningful in the corneal healing context?
Minor language polishing is recommended.

Reviewer 2 Report

Comments and Suggestions for Authors

The paper has good data but the interpretation is overstated. Results need more context (comparisons, benchmarks), Discussion should move from description to critical analysis, and the Conclusion must be toned down to reflect preliminary, in vitro evidence only. My observations are listed below:

For MTT assay, authors only tested 0.01–0.1 mg/mL (line 165). Why this narrow range? No dose–response curve beyond that.

 No positive control for proliferation (e.g., EGF) and no cytotoxic control (e.g., H₂O₂).
The “optimal concentration 0.08 mg/mL” (line 229) looks arbitrary since it is based only on 72h readout. A true dose–response curve across timepoints would strengthen the claim.

At scratch assay only untreated vs. treated groups tested. A positive control (e.g., growth factor, EGF) would validate the model. Without it, conclusions about wound-healing potential are limited.

Data are presented clearly, but several statements are purely descriptive without context. For example, “33.8% DPPH radical scavenging activity” (line 222) is reported, but no comparison with a standard antioxidant (like BHA or ascorbic acid) is made. Readers don’t know if 33.8% is strong, moderate, or weak.

Figure legends are overloaded with redundant statistical notes (Figures 3–4). The use of multiple symbols (#, @, ©, $) is confusing and hard to follow. A simpler annotation system (letters a, b, c) would improve clarity.

In Figure 2 (cell viability), the authors claim 0.08 mg/mL as “optimal,” but the figure only shows a narrow dose range. Without a broader curve, the claim looks overstated.

The Discussion section is too descriptive and feels short on critical analysis. Much of it repeats results or lists compounds without deeper interpretation. For a high-impact journal, the section should go beyond description and address mechanisms, clinical translation, and limitations. For example:

You could discuss how flavonoids/anthocyanins might influence wound-healing pathways

Several comparisons are superficial (e.g., “anthocyanin levels higher than reported elsewhere” lines 321–323). Authors should address why extraction differences matter biologically.

Some interpretations sound speculative. Example: “balanced upregulation observed in this study reinforces CTE’s therapeutic potential” (lines 350–351). The term balanced is not defined, and no direct functional assay supports this claim.

Future directions are missing: testing in primary corneal cells, animal models, and formulation development for ocular delivery.

Current conclusion is too strong: “CTE promotes corneal epithelial wound healing” (line 355–359). The data are in vitro only; thus, wording should be toned down to “CTE shows potential to support corneal epithelial repair in vitro.”

No mention of limitations in the conclusion — a high-impact journal would expect at least a sentence acknowledging the need for in vivo validation.

Reviewer 3 Report

Comments and Suggestions for Authors

The topic of the study, namely the search for substances with therapeutic potential for corneal abrasion, appears to be well-justified. The paper addresses the effects of an extract from fresh Clitoria ternatea flowers. The work requires corrections. Below are my comments and suggestions:

Comment 1: Defining the plant material from which the extract was obtained requires additional information. Providing the name of the town where the plant material was purchased is not sufficient. The name of the plant material producer or the location of cultivation (coordinates) should also be provided. The material must be defined to allow for reproducible identification. Information on how the authentication was performed and who performed it should also be included.

Comment 2: The Authors used an LC-MS database to identify the compounds. It is important to specify the exact database. Given that no standard substances were used and the identification was based solely on MS data without fragmentation, the identification of the substance in the extract is very rough and subject to error. This is especially true since the paper does not provide information on whether the identification was also based on data from other authors regarding this species, or data on other species related to the one studied. This issue should be addressed in the paper. Furthermore, only a rough profile was determined, and no quantitative determination of the substance was performed. Therefore, it is not certain which compounds dominate and what their content is, making it difficult to relate this to the observed biological activity. It is not known whether another raw material purchased from a store will have a similar quantitative profile. This information should be included in the limitations of the study..

Comment 3: Another issue is the identification of the substance: an aqueous extract was prepared at 50°C. Please explain the presence of lipophilic compounds that are poorly soluble in water, such as fatty acids, steroids, and terpenoids.

Comment 4: line 340-352. In my opinion, the conclusions drawn are incorrect. Even if CK3 or Cx43 expression was elevated at 12 hours, this finding was not statistically significant compared to the untreated group, which served as a control (values also increased in this untreated group; the authors themselves indicate this in lines 266-277). Therefore, it is impossible to draw conclusions about a positive effect of the analyzed extract in these specific analyses. This paragraph and lines 357-359 should be reworded and reconsidered. The authors should always refer to the control at a given time point, not to the change over time vs. time "0", given that values also increased in the control group.

Comment 5: Line 94 - "DNA/RNA-free distilled water" was used for extraction - the source of the water should be specified: purchase/company or apparatus used to obtain the water.

Comment 6: Line 107: There is information only about ESI-, and the article also includes ESI+. Add the appropriate information.

Comment 7: the source of the cell lines used in the study should be specified

Comment 8: The abbreviation "CTE" is used to refer to several different things: 1. the plant species (e.g., lines 78-84), 2. the fresh flowers of the species (line 92), and 3. the flower extract of the plant (e.g., lines 14). The entire manuscript should be reviewed and appropriately modified to differentiate.

Comment 9: Line 363 -364. The description of the contents of Tables S1 and S2 does not correspond to reality.

Comment 10. The caption of Figure 1 contains the words "Graph A" and "Graph B," which are not present in the figure. The figure or caption should be modified accordingly.

Round 2

Reviewer 2 Report

Comments and Suggestions for Authors

The authors have addressed the reviewers’ comments and implemented the suggested improvements. In my opinion, the paper is now suitable for acceptance and publication.

Reviewer 3 Report

Comments and Suggestions for Authors

The manuscript has been sufficiently improved. In my opinion, it can be accepted for publication in its current form.